# Effectiveness of the Research Practice Ability Enhancement Program on Evidence-Based Practice Competencies in Clinical Nurses: A Non-Randomized Controlled Trial

**DOI:** 10.3390/healthcare13182273

**Published:** 2025-09-11

**Authors:** Sun-Ae Kim, Hye-Won Jeong

**Affiliations:** Department of Nursing, Korea National University of Transportation, 61, Daehak-ro, Yonggang-ri, Jeungpyeong-eup, Jeungpyeong-gun 27909, Chungcheongbuk-do, Republic of Korea; sakim@ut.ac.kr

**Keywords:** evidence-based practice, Republic of Korea, nurses, nurse educators, nursing education research

## Abstract

Background/Objectives: Evidence-based practice (EBP) remains limited among clinical nurses worldwide, with Korean healthcare settings facing challenges. This study examined the effectiveness of the research practice ability enhancement program (RPAEP) in strengthening evidence-based practice (EBP) competencies among clinical nurses who had completed master’s coursework without a thesis. Methods: A non-randomized controlled trial was conducted at a tertiary hospital in Korea (June–December 2022). Thirty participants were assigned by convenience sampling to intervention (n = 15) or control (n = 15) groups. The intervention comprised 12 biweekly sessions. The primary outcome was research practice ability (RPA); the secondary outcomes were EBP beliefs (EBPBs), EBP Attitudes (EBPAs), and Nursing Professional Values (NPVs). Analyses employed Wilcoxon signed-rank and Mann–Whitney U tests. The qualitative evaluation used thematic analysis of focus group interviews (FGIs) (n = 12). Results: All participants completed the study. The intervention group showed significant improvements in RPA (within-group: Z = −1.96, *p* = 0.050, ES = 0.82; between-group: t = −2.39, *p* = 0.016, ES = 1.02) and EBPBs (t = −3.30, *p* = 0.005, ES = 0.87). NPVs showed significant between-group differences (t = 2.38, *p* = 0.024, ES = 0.87), while EBPAs remained unchanged. The FGIs revealed three major themes related to participation in the research practice ability enhancement program: “barriers to research practice,” “guidance for research practice,” and “enhancing research practice ability.” Conclusions: The RPAEP enhanced nurses’ EBP competencies despite the non-randomized design. However, single-site implementation and convenience sampling limit generalizability. Sustained EBP integration requires addressing individual and organizational barriers through comprehensive education with institutional commitment.

## 1. Introduction

Evidence-based practice (EBP) involves the integration of the best available evidence from high-quality research with clinical expertise and patient values to improve healthcare outcomes [1]. Implementing EBP requires nurses to develop competencies in research methodologies, critical appraisal, and the application of scientific findings in clinical practice [2]. Despite its recognized importance, many clinical nurses encounter difficulties in conducting research and applying EBP in daily practice [3]. Barriers such as limited research training, lack of systematic literature review exposure, and time constraints due to heavy workloads hinder research participation [4]. Additionally, institutional factors such as inadequate organizational support and an absence of structured mentorship further restrict nurses’ ability to engage in meaningful research activities [5].

The nature of these barriers varies across healthcare contexts and organizational cultures [6]. In many cases, clinical nurses rely on experiential knowledge, peer observations, and traditional practices rather than incorporating the latest research evidence into their decision-making processes [7]. This reliance on non-empirical sources of knowledge persists even when evidence-based alternatives are available, suggesting that barriers extend beyond mere resource limitations to encompass deeply ingrained professional practices and cultural norms [6]. Organizational challenges, such as inadequate institutional support and the absence of structured mentorship, further restrict nurses’ ability to engage in meaningful research activities [8].

The limited availability of undergraduate research training in Korea exacerbates these barriers [8]. Historically, Korean nursing programs have prioritized theoretical education over research-based learning, and until 2005, undergraduate curricula did not formally include EBP training [8]. Although recent curriculum reforms have integrated some aspects of research training, most nurses are first introduced to research methodologies at the graduate level [9]. Furthermore, while some hospitals provide continuing education programs on EBP, systematic research training remains insufficient, leading to significant variability in nurses’ research competencies [1]. Consequently, many practicing nurses are unfamiliar with EBP methodologies and are first introduced to research concepts at the graduate level [7]. This gap underscores the necessity of structured educational interventions that equip nurses with the knowledge and skills required for EBP implementation [5].

Addressing the barriers to EBP adoption requires a systematic approach that integrates research training into clinical nursing education [10]. The implementation of mentorship-driven programs provides an effective strategy for equipping nurses with research capabilities [10]. While previous studies have extensively examined the effectiveness of mentorship-based educational programs in nursing, limited research has explored their applicability in Korea’s unique healthcare system [11]. Unlike Western healthcare settings, where mentorship is often well-structured within hospitals, Korean clinical environments present hierarchical challenges that may inhibit junior nurses from engaging in research [12]. This study contributes to the existing literature by developing and evaluating a structured research practice ability enhancement program (RPAEP) specifically tailored for clinical nurses in Korea. Unlike previous interventions, RPAEP integrates personalized mentorship, structured research workshops, and digital tools to support research engagement in hospital settings. Research mentors play a pivotal role in facilitating the acquisition of knowledge, reinforcing the importance of scientific inquiry, and fostering a culture that encourages evidence-based decision making [13]. By receiving structured guidance, nurses can develop confidence in formulating research questions, conducting systematic literature reviews, and applying research findings to their practice [14].

The Advancing Research and Clinical Practice through Close Collaboration (ARCC) model provides a conceptual framework for understanding the facilitators and barriers to EBP implementation [1]. This model highlights the significance of organizational culture, mentorship, and training programs in overcoming research challenges [15]. According to the ARCC model, the successful adoption of EBP requires healthcare institutions to identify organizational strengths and barriers, assess institutional readiness for EBP, and implement mentorship strategies that reinforce research competencies among nursing staff [1].

The Korean healthcare system presents unique challenges that impact the integration of EBP within nursing practice [16]. A hierarchical workplace structure often limits opportunities for junior nurses to engage in research, as clinical decision making is frequently guided by senior practitioners rather than empirical evidence [17]. Additionally, the absence of dedicated research programs within many hospital settings limits the exposure of nurses to scientific inquiry and critical appraisal processes [18]. These challenges necessitate the development of targeted educational interventions that not only provide research training but also create an institutional culture that values evidence-based decision making. In addition, prior evidence indicates that nurses’ professional roles have expanded to include prescribing activities, which are associated with both challenges and positive outcomes [19]. These developments underscore the need to strengthen nurses’ competencies in generating and using evidence.

To bridge the gap between research knowledge and clinical application, this study developed and assessed the effectiveness of a research practice ability enhancement program (RPAEP). This program was designed to provide structured mentorship, research training, and hands-on learning experiences to clinical nurses, enabling them to engage in EBP more effectively. The research objectives were formulated to evaluate the impact of the RPAEP on nurses’ research competencies and to identify the changes that occurred in their clinical practice following program participation. The study was guided by the following research questions:How does the RPAEP affect the research practice abilities of clinical nurses?What changes have occurred in the practice of clinical nurses after participating in the RPAEP?Based on these research questions, we formulated the following hypotheses:Primary hypothesis:The RPAEP will significantly improve research practice ability (RPA) scores in the intervention group compared to those in the control group.Secondary hypotheses:The intervention group will demonstrate significantly higher EBP belief (EBPB) scores compared to the control group following the RPAEP.The intervention group will show significantly improved EBP Attitude (EBPA) scores compared to the control group.The intervention group will exhibit significantly enhanced Nursing Professional Value Scale (NPVS) scores compared to the control group.

Additionally, qualitative exploration will identify barriers, facilitators, and perceived changes in clinical practice following program participation.

By addressing these questions, this study contributes to the growing body of literature on mentorship-driven research training in nursing and provides insights into how structured educational programs can enhance EBP engagement in clinical settings.

## 2. Materials and Methods

### 2.1. Study Design

This study used a mixed design to evaluate the effect of an RPAEP led by educational nurses on clinical nurses. To evaluate the effects of the RPAEP, a quantitative quasi-experimental design with a non-equivalent control group pre-test/post-test approach was used. Additionally, a qualitative content analysis of focus group interviews was conducted after the program to investigate changes in the clinical practice of the participating nurses. This study is an expanded version of preliminary findings that were presented in abstract form at the 2024 Korean Society of Nursing Science Fall Conference. The full text has not been published elsewhere [20].

### 2.2. Setting and Sample

The participants were nurses employed at C University Hospital in G Metropolitan City, Korea, who had completed master’s coursework without a thesis submission. A total of 30 nurses meeting these criteria were recruited using convenience sampling. Allocation was not randomized. Instead, participants were assigned to the experimental (n = 15) or control (n = 15) groups according to their voluntary willingness either to participate in the intervention or to serve as controls. Sample size was calculated using G*Power 3.1.9.4. With α = 0.05, power = 0.80, and a large effect size, the minimum required sample size was 24. We based our assumption of a large effect size on previous intervention studies targeting clinical nurses’ evidence-based practice competencies. Kim and Jeong [21] reported substantial improvements in research practice ability and EBP beliefs, with standardized mean changes exceeding d = 1.2. Given this precedent of large effects, we judged a minimum of 24 participants as sufficient. Given a significance level of α = 0.05 and an effect size of d = 1.2, a minimum of 26 participants was required to maintain the test power (1 − ß = 0.80). Considering the dropout rate, a total of 30 participants—15 per group—were recruited. All 30 nurses remained in the study until completion, with no attrition. All participants completed both baseline and post-test questionnaires, and there were no missing data. After the program ended, 12 participants completed focus group interviews, which involved three groups of four individuals.

### 2.3. Ethical Consideration

This study was approved by the IRB (IRB no. CNUH-2022-124) at C University Hospital in G Metropolitan City, Korea. Participants were provided with explanations of the research and consent forms containing information on the study’s purpose, data collection procedures, and the confidentiality of personal information. They were informed that participation would not involve any disadvantages and that they could withdraw from the study at any time.

### 2.4. Measurements

#### 2.4.1. RPA

RPA was measured using the “research practice” section from the Program Outcome Self-Assessment Tool in Korean Nursing Baccalaureate Education developed by Kim [22] for nursing students. The tool was modified and supplemented to fit the nursing practice context. The tool includes seven items, each rated on a 4-point Likert scale (1 = very difficult, 4 = very easy). The higher the score, the higher the RPA. Kim’s [22] initial study reported that Cronbach’s α = 0.86. In the present study, Cronbach’s α was 0.92.

#### 2.4.2. EBPB

To measure EBPB, the Korean version of the EBPB scale developed by Melnyk et al. [23] was used. This tool consists of 16 questions and measures the level of support for EBP and confidence in one’s ability to engage in EBP. Items were rated on a 5-point Likert scale (1 = very negative, 5 = very positive); the higher the score, the stronger the EBPB. The reliability of the original instrument [23] was supported where Cronbach’s α = 0.85. In the present study, Cronbach’s α was 0.84.

#### 2.4.3. EBPA

To measure EBPA, the Korean version of the EBPA scale developed by Aarons [24] was used. This tool consists of four questions on openness, four questions on appeal, four questions on divergence, and three questions on requirements. Items were rated on a 5-point Likert scale (0 = not at all, 4 = very much); the higher the score, the more positive the EBPA. Aarons’ [24] original validation reported a Cronbach’s α of 0.96. In the present study, Cronbach’s α was 0.81.

#### 2.4.4. NPVs

Nursing professionalism was measured using the NPVs developed by Yeun et al. [25]. This tool consists of 29 items across five sub-areas: professional self-concept (9 items), social awareness (8), nursing expertise (5), nursing practice (4), and nursing independence (3). Items were rated on a 5-point Likert scale (1 = strongly disagree, 5 = strongly agree); the higher the score, the higher the level of nursing professionalism. The original validation [25] showed a Cronbach’s α of 0.92. In the present study, Cronbach’s α was 0.95.

### 2.5. Procedure

#### 2.5.1. Development of the RPAEP

The RPAEP was developed based on the ARCC model, which provides a systematic framework for implementing EBP in healthcare settings. The development process began with a comprehensive literature review to identify existing EBP education programs and their effectiveness in clinical settings. This review revealed that successful programs typically incorporated mentorship components, hands-on practice opportunities, and ongoing support structures.

Following the literature review, an expert panel consisting of clinical nurse educators, research methodologists, and EBP specialists was convened to adapt the ARCC model’s core components to the Korean healthcare context. The panel focused particularly on addressing the unique challenges faced by Korean nurses, including hierarchical organizational structures and limited exposure to research methodology during undergraduate education. Through iterative discussions and reviews, the panel developed preliminary program content that integrated adult learning principles with cultural considerations specific to Korean healthcare settings.

The initial program structure was validated through a pilot study conducted with 11 clinical nurses at a research hospital between September and November 2021. The pilot program consisted of 8 sessions and provided valuable insights into the practical challenges of implementing research education in clinical settings [12]. Participant feedback highlighted the need for additional support in research topic selection, methodology choice, and paper writing. This feedback led to the expansion of the program to 12 sessions, with enhanced focus on these specific areas of difficulty (Figure 1).

An educational needs assessment conducted with the participants before the program revealed specific challenges in the research process. Participants reported experiencing delays in writing their papers due to difficulties in selecting appropriate research topics, choosing suitable methods, and managing the writing process effectively. In response to these identified needs, the program was structured to provide comprehensive support in literature review techniques, research topic development, and academic writing skills.

The final program structure incorporated three key components: theoretical foundation, practical application, and mentorship integration. The theoretical component included instruction in research methodology principles, EBP frameworks, and critical appraisal skills. Practical application sessions focused on hands-on experience with research tools, including database searching, reference management software (EndNote), and statistical analysis tools (G*Power). The mentorship component provided individualized guidance through research project development and implementation.

To facilitate continuous learning and communication, the researchers established an online platform where individual research notebooks were created for each participant. These digital notebooks served as repositories for session reviews, research ideas, and questions, enabling ongoing dialogue between participants and mentors. The program utilized a blended learning approach, combining face-to-face workshops with online learning modules and individual mentorship sessions.

All educational materials were developed in the Korean language and incorporated local healthcare examples to ensure relevance and applicability. The materials underwent multiple rounds of review and refinement based on expert feedback and pilot program outcomes. Special attention was paid to creating resources that would support participants in overcoming common barriers to research implementation in Korean healthcare settings.

Two clinical nurse educators with extensive research experience served as EBP mentors throughout the program. These mentors played crucial roles in providing individualized guidance, facilitating group discussions, and supporting participants in applying research concepts to their clinical practice. The mentorship component was designed to create a supportive learning environment while respecting traditional hierarchical structures within Korean healthcare institutions.

#### 2.5.2. Application of the RPAEP

The program was implemented from June to December 2022. The sessions were held at two-week intervals in the hospital’s education room. The researchers served as EBP mentors. All participants used their personal laptops during the sessions. After completing each session, the participants independently reviewed its contents for two weeks. The researchers, acting as mentors, created individual research notebooks for the participants using an online platform. To encourage the nurses to constantly communicate with their mentors to enhance their research capabilities, the nurses were instructed to record their review content and research ideas and write down any questions in their notebooks after each session (Figure 2). Meanwhile, the control group did not receive any intervention during the study period. They only completed the baseline and post-test questionnaires, and no access to the intervention content was provided prior to the post-test. At the time of the postsurvey, they were provided with a booklet summarizing the educational program for information purposes. Nurses in the control group who wished to participate in the full program were later offered the RPAEP in 2023; however, outcomes from this later participation were not assessed as part of the present study.

### 2.6. Data Collection

#### 2.6.1. Quantitative Data Collection

The study recruited clinical nurses who had completed a master’s degree program without writing a master’s thesis over a two-week period in May 2022. All participants expressed their intention to participate in the study and provided informed written consent. Data were collected from June 2022 to December 2022. For quantitative data, a pre-survey was administered to the intervention group immediately before the program’s initiation, and a post-survey was conducted immediately upon its conclusion. A research assistant collected the completed surveys.

#### 2.6.2. Qualitative Data Collection

After the RPAEP, qualitative data were collected only from nurses in the intervention group using a focus group method. Three focus groups were organized, each with four participants (n = 12), to ensure both diversity of perspectives and active engagement. The sessions were facilitated by the corresponding author, who also served as a clinical nurse educator during the intervention. The facilitator had formal training and prior experience in qualitative research methods and interview techniques. To minimize potential bias arising from this dual role, a validated semi-structured guide was used, professional boundaries were maintained, and the data analysis was cross-checked by another researcher. Each focus group was conducted in January 2023, lasted approximately 60 min, and data collection continued until no new themes emerged, indicating data saturation. The interviews were recorded, and the researchers took field notes of the participants’ nonverbal expressions. The key questions were as follows: “What barriers did you encounter when conducting research? What positive and negative experiences did you have with program? What do you think the program helped you achieve?” Each interview lasted approximately 60 min. The interviews were held in a quiet conference room at the hospital, scheduled during the participants’ convenient times to minimize external interferences. They were conducted at various times of the day, based on participants’ availability, to accommodate their work schedules and personal commitments. The researcher ensured a comfortable environment to facilitate open and honest discussions. Focus groups were chosen to facilitate interactive discussion and the sharing of experiences among participants. Themes identified across the three focus groups were compared to ensure consistency, and no discrepancies emerged. This qualitative approach enabled us to capture context-specific barriers and facilitators that could not have been sufficiently identified through systematic reviews, thereby complementing the quantitative results with richer, first-hand insights.

The researcher played a crucial role in data collection and had received training in qualitative research methods and interview techniques. The researcher had experience conducting qualitative research several times, ensuring familiarity with effective data collection practices. Their relationship with the participants was professional, with no prior personal connections, ensuring objectivity. The validity of the interview questions was established through expert review and pilot testing.

### 2.7. Data Analysis

The collected quantitative data were analyzed using SPSS/WIN 25.0 (SPSS Inc., Chicago, IL, USA). The statistical significance level was α = 0.05. The reliability of the measuring tool was tested using Cronbach’s α. Descriptive statistics were used to identify the general characteristics of the EBPB, EBPA, NPV, and RPA scales. Normality was examined for all outcome variables. The pre-test RPA scores did not satisfy the normality assumption; therefore, within-group differences were analyzed using the Wilcoxon signed-rank test. The post-test RPA scores satisfied the normality assumption, so independent t-tests were used for between-group comparisons. For the other variables (EBPB, EBPA, NPVS), paired and independent t-tests were applied as appropriate. Effect sizes (ESs) with 95% confidence intervals were also calculated to estimate the magnitude of intervention effects.

The collected qualitative data were stored in the NVivo 12 program (QSR International, Burlington, MA, USA), and classification and coding were performed according to semantic units. We followed Hsieh and Shannon’s [26] conventional qualitative content analysis. All focus group recordings were transcribed verbatim and de-identified. Two researchers independently read the transcripts several times to achieve immersion and highlighted meaning units relevant to the research questions. Using line-by-line open coding, initial codes were inductively generated from participants’ narratives. Through constant comparison, similar codes were grouped into subcategories and further organized into higher-order categories. A preliminary codebook was developed after coding the first two transcripts and refined iteratively across subsequent transcripts. Discrepancies were resolved through discussion, and when necessary, a third reviewer adjudicated disagreements to ensure reliability.

## 3. Results

### 3.1. General Characteristics of the Participants

Table 1 shows the participants’ general characteristics. The average age of the experimental group was 40.5 years, while the control group had an average age of 38.4 years, with no significant difference between the two groups. Similarly, clinical experience was homogeneous between the groups, with the experimental group averaging 17.9 years and the control group 15.3 years. There were also no significant differences between the groups in terms of work unit and the duration since completing a postgraduate degree. At baseline, both groups showed no significant differences in their RPA, EBPB, EBPA, or NPVS.

### 3.2. Comparison of Research Variables Between Experimental and Control Groups in the RPAEP

For RPA, the experimental group experienced a significant improvement between the pre- and post-test scores (Z = −1.96, *p* = 0.050, ES = 0.82, 95% CI: 0.07–1.56), and a significant difference was observed between the groups at the post-test (t = −2.39, *p* = 0.016, ES = 1.02, 95% CI: 0.26–1.79). The experimental group also demonstrated a significant increase in EBPB from pre-test to post-test (t = −3.30, *p* = 0.005, ES = 0.87, 95% CI: 0.13–1.62), whereas the control group showed a non-significant increase. Regarding NPVS, there was no significant change in pre- and post-test scores within either group, although the post-test scores of the experimental group were significantly higher than those of the control group (t = 2.38, *p* = 0.024, ES = 0.87, 95% CI: 0.12–1.62). In contrast, there was no significant difference in EBPA between the two groups (Table 2).

### 3.3. Barriers to Research and Changes Based on Participation in the RPAEP

The qualitative findings revealed three overarching themes: barriers to research practice, guidance for research practice, and enhancing research practice ability.

#### 3.3.1. Theme 1. Barriers to Research Practice

Participants described a variety of barriers that hindered their ability to initiate or sustain research.

*Situational factors* such as pregnancy, childbirth, parental leave, and departmental transfers disrupted continuity in research planning. One participant stated, “*While I was completing my master’s course, I couldn’t start research right away due to pregnancy, childbirth, and parental leave*” (Participant 2). Another recalled, “*After completing the master’s course and deciding on my thesis topic, I had no time to conduct research due to moving departments and adapting to my new department*” (Participant 3). In some cases, changes in supervision added further difficulty: “*Due to the retirement of my advisor, I lost my research direction*” (Participant 5).

Participants also reported *difficulty and anxiety in initiating research*. For example, one said, “*Six years after completing my master’s course, I was embarrassed because I couldn’t remember any research methods, such as how to review literature and use EndNote*” (Participant 1). Others expressed self-doubt and lack of confidence: “*It is too difficult to review the literature, and I have no confidence in whether I will be able to do the research*” (Participant 2). Some described negative past experiences: “*I tried to write a thesis while completing my master’s course, but I failed. I forgot all the methods for conducting research 10 years later, so it was a burden to try to initiate research alone*” (Participant 7). Another shared, “*After completing my master’s course, I wanted to take a break and start research, but I couldn’t really afford it*” (Participant 8).

A further challenge was *difficulty in selecting research topics*. One participant noted, “*Every time there was a block in the process of finding a research topic, I lost my will and gave up*” (Participant 9). Another admitted, “*I wanted to find a thesis topic in my field, but I couldn’t figure out what to do*” (Participant 10). For some, not completing a thesis during their master’s program remained a burden: “*Graduation was a burden on my heart because I did not write a thesis after completing my master’s course, and the process of conducting research was not easy because it was hard to select a research topic by myself*” (Participant 12).

These accounts illustrate how personal circumstances, skill decay, psychological barriers, and difficulties in topic selection interacted to prevent sustained research engagement.

#### 3.3.2. Theme 2. Guidance for Research Practice

Despite these challenges, participants highlighted the importance of structured support and mentorship.

Under *assistance and support from mentors*, one nurse reflected, “*Through the mentor, I was able to learn that research, which I only thought was difficult, can grow out of experience and work in the field. In particular, it was very helpful for me when the mentor explained actual research cases*” (Participant 4). Another explained, “*Unlike nursing college professors, my mentor, who worked in the clinical field and had rich research experience, suggested feasible research topics and methods, so I was able to set the direction for my research plan. I was glad to have confidence that I could do it*” (Participant 5). Similarly, a participant appreciated personal guidance: “*The mentor gave an easy-to-understand lecture on how to conduct research and provided lecture materials, which was very helpful when reviewing at home. Also, I lost my direction after my advisor retired, so I was grateful that he connected me with a new advisor*” (Participant 11).

The value of a *systematic educational program* was also emphasized. One nurse said, “*It was very helpful in everything from selecting a research topic to suggesting a direction for research and systematically teaching how to search for high-quality journals. I learned how to use EndNote and G*Power for the first time, and it was very helpful when writing a research plan*” (Participant 6). Another explained, “*Thanks to step-by-step programs for finding key questions for research, conducting a systematic literature review, and research design methods, I was able to try everything and write a research plan*” (Participant 9). Others highlighted how structured training in literature search skills improved their efficiency: “*Even 10 years before I completed my master’s degree, there were RISS and PubMed, but at that time I did not know how to conduct a search, so I searched randomly. This program helped me learn how to search journals effectively and efficiently step by step*” (Participant 12).

Collectively, these findings underscore that individualized mentoring and systematic educational approaches were critical in enabling participants to overcome barriers and progress in research practice.

#### 3.3.3. Theme 3. Enhancing Research Practice Ability

Participants described how the program enhanced their confidence, competence, and professional commitment to evidence-based practice.

In terms of *improving confidence in research practice and EBP ability*, one nurse shared, *“While receiving education with nurses with similar circumstances to mine, I gained the confidence that I could do research, and after writing even one line of a research plan myself, I found joy in learning about research”* (Participant 1). Another reflected on a mindset shift: *“Over time, my perspective shifted, and I developed the motivation to pursue research actively rather than postponing it. I became excited and confident that I could escape the completion of my master’s course. In addition, when I needed evidence at work, I began conducting literature searches, and I grew into a person who ponders and thinks about evidence”* (Participant 8). Others emphasized skill improvement: *“My ability to search the literature improved, and I was able to find the contents I was curious about. So I was able to perform evidence-based work and, furthermore, I was able to find a thesis topic after thinking about various topics”* (Participant 11).

Participants also described *enhancing healthcare quality through evidence-based practice*. For instance, *“As the research capacity of nurses improves, efforts are being made to provide evidence-based care for nursing patients, which improves the quality of nursing, including patient services”* (Participant 2). Another explained, *“Understanding the latest nursing research trends and discovering how they differ from current understandings in nursing work can inspire change and improve nursing professionalism”* (Participant 4). A third participant highlighted the broader cultural change: *“The number of nurses who look at the clinical field from a new perspective and research to improve the quality of medical care will increase, and a culture of applying evidence-based practice within medical institutions will be created, which will help improve patient safety”* (Participant 9).

These narratives demonstrate how the program not only improved individual confidence and skills but also motivated participants to contribute to organizational and patient care improvements through evidence-based practice.

## 4. Discussion

This study analyzed how the RPAEP affected clinical nurses and the research barriers they faced. Beyond individual skill development, the program demonstrated the potential for structured educational interventions to transform research culture in clinical settings. The program enhanced beliefs about EBP in the experimental group, signifying not only improved understanding but also a fundamental shift in how nurses perceive the role of research in their practice. A low belief in EBP signifies poor understanding of its benefits for healthcare quality and patient outcomes [27]. Because insufficient knowledge about EBP is a major barrier, organizational-level educational programs are necessary to foster knowledge and positive beliefs about EBP [28].

While the program successfully enhanced understanding and engagement with EBP, the findings revealed a complex interaction between individual learning and organizational context. Despite positive beliefs about EBP, barriers such as individual time constraints and poor organizational support may limit EBP implementation [29]. This highlights the need for a comprehensive approach that addresses both individual capacity building and systemic organizational change [30]. The success of EBP implementation appears to depend on creating an ecosystem where research skills can be effectively applied and sustained [31].

The program significantly increased professional commitment in the experimental group compared to the control group; this aligns with previous research indicating that EBP can reduce self-efficacy and, relatedly, burnout; increase job satisfaction; and improve professionalism among nurses [32,33,34,35,36]. The dynamic learning process observed through the focus group interviews revealed how participants progressed from initial uncertainty to active engagement with research. This progression was characterized by increasing confidence in questioning existing practices and seeking evidence-based alternatives [37]. The participants demonstrated growing professional autonomy by actively identifying how the latest evidence diverged from their practice and understanding recent nursing research trends [35].

The mentorship component emerged as a crucial catalyst for learning and professional growth. The focus group interviews revealed that mentors did more than simply transfer knowledge; they facilitated a dynamic learning environment where nurses could safely explore new ideas, challenge existing practices, and develop their research identity [38]. This supportive relationship helped participants build confidence in their research abilities while creating an environment conducive to learning and innovation [29].

Regarding barriers, the participants identified situational factors, difficulty, and anxiety in starting their research, and difficulty in selecting research topics, which have also been reported in other studies [35,39]. However, the program demonstrated that these barriers can be effectively addressed through structured support and guidance. The interactive nature of the learning process, where participants could discuss challenges and solutions with mentors and peers, proved particularly valuable in overcoming these obstacles.

The study findings indicate that structured educational programs like the RPAEP may strengthen nurses’ engagement in research and evidence-based practices by building confidence and competence. Several methodological limitations should be acknowledged. This study was conducted in a single university hospital with a small sample, which constrains the generalizability of the findings. The use of convenience sampling for both the intervention and control groups may have introduced selection bias, thereby limiting representativeness. Participants were restricted to nurses who had completed master’s coursework without thesis submission; while this criterion ensured a comparable educational background, variations in time since coursework completion and individual academic experiences could not be controlled. These differences may have influenced baseline competencies and intervention outcomes. The results may also reflect organizational and cultural characteristics specific to Korean nursing practice, which could reduce applicability to other settings. Another limitation is that the researcher who facilitated the intervention also conducted the focus groups, which may have introduced bias in participants’ responses. Although this risk was mitigated through the use of a validated semi-structured guide, maintenance of professional boundaries, and the cross-checking of the data analysis by another researcher, the potential for bias due to this dual role cannot be fully excluded. Finally, the power analysis in this study was based on a relatively large expected effect size (d = 1.2), which reduced the required sample size. Although this assumption was informed by previous studies, it may not adequately capture more modest yet meaningful intervention effects. As such, the reliance on a very large effect size increases the possibility of a Type II error, and future studies should consider more conservative assumptions when estimating sample sizes. The program also required dedicated mentors, multiple face-to-face sessions, and institutional support, which may limit implementation in resource-constrained environments.

Beyond methodological considerations, the study provides practical implications for expanding similar programs within Korean hospital settings. Successful expansion will depend on supportive policy measures, such as providing protected time for nurses to participate in research, offering recognition and incentives linked to professional development or remuneration, and ensuring adequate staffing levels to alleviate workload. Relational and organizational contexts must also be considered, as hierarchical structures may discourage active participation. Furthermore, modern AI and digital platforms hold potential to support scalability by enabling adaptive online learning, virtual mentoring, and automated assistance with literature searches and data analysis. For nurse educators, these findings underscore the importance of integrating structured mentorship with digital resources into professional development initiatives, thereby fostering a sustainable culture of evidence-based practice in clinical settings.

Future research should build upon these findings by testing modified delivery formats—such as online modules, peer mentoring, or condensed curricula—that may reduce resource demands while retaining educational effectiveness. Multisite studies with larger and more diverse samples are needed to enhance generalizability, and longitudinal designs could determine whether improvements in research practice translate into sustained professional growth, organizational change, and improved patient outcomes. In addition, economic evaluations that compare program costs with outcomes such as nurse retention and quality indicators would provide valuable evidence for decision-makers considering broader implementation.

## 5. Conclusions

This study demonstrated the effectiveness of the RPAEP in improving beliefs in EBP and research practice abilities among clinical nurses. Through mentorship and structured education, the program successfully enhanced nurses’ confidence and competence in research practice. The findings indicate that organizational support and resources are essential to facilitate EBP implementation, and healthcare institutions should foster cultures that encourage clinical nurses to actively engage in research activities. However, the findings also emphasize that organizational support and resources are necessary to nurture EBP implementation. Healthcare institutions should focus on fostering organizational cultures that encourage clinical nurses to engage in EBP while providing necessary resources and support systems. A comprehensive approach combining education, mentorship, and organizational support is essential to improve healthcare quality and patient outcomes through evidence-based nursing practice. In conclusion, developing research capabilities among clinical nurses requires a multi-faceted approach that combines structured education, mentorship, and organizational resources. Given the small sample size and single-site design, the results should be interpreted with caution, and further research across diverse settings is warranted to confirm broader applicability.

## Figures and Tables

**Figure 1 healthcare-13-02273-f001:**
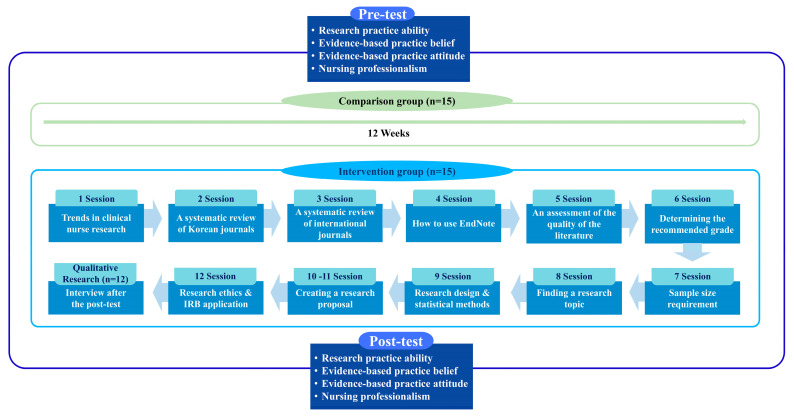
The research practice ability enhancement program based on the Advancing Research and Clinical Practice through Close Collaboration model.

**Figure 2 healthcare-13-02273-f002:**
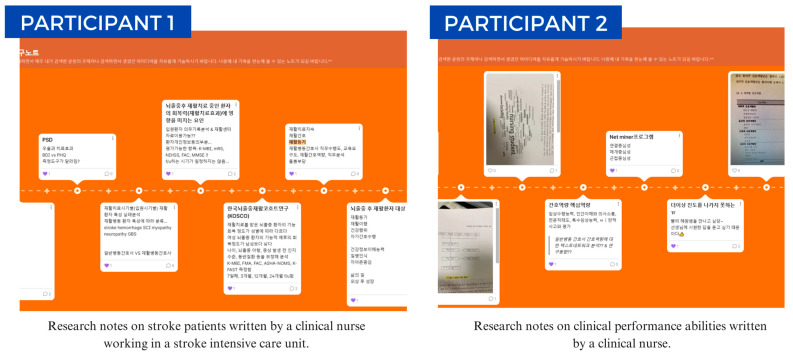
The research notes of the participants. (Participant 1 documented topics related to stroke patient care and rehabilitation factors (Factors influencing post-stroke rehabilitation outcomes; Continuation of rehabilitation treatment; Korean Stroke Cohort for Outcomes in Rehabilitation, KOSCO; Rehabilitation factors for post-stroke patients; Post-stroke depression). Participant 2 documented aspects of nursing competencies and performance (Core competencies in nursing practice; Reasons for difficulty in progressing with daily rounds; Creativity, competency, and autonomy)).

**Table 1 healthcare-13-02273-t001:** Baseline general characteristics of the participants.

Characteristics	Categories	Total (n = 30)	Experimental (n = 15)	Control (n = 15)	χ^2^ or t	*p*
		n (%) or	n (%)	n (%)		
Age (years)	<40	12(40.0)	5(33.3)	7(46.7)	0.56	0.710
	≥40	18(60.0)	10(66.7)	8(53.3)		
	Mean ± SD	39.47 ± 5.43	40.53 ± 5.24	38.40 ± 5.58	1.08	0.289
Total working career (years)	<10	3(10.0)	0(0)	3(10.0)	4.87	0.088
	≥10~<20	17(56.7)	11(73.3)	6(20.0)		
	≥20	10(33.3)	4(26.7)	6(20.0)		
	Mean ± SD	16.60 ± 4.98	17.87 ± 4.69	15.33 ± 5.09	1.42	0.167
Work unit	Ward	11(36.7)	5(33.3)	6(40.0)	0.20	0.977
	ICU	4(13.3)	2(13.4)	2(13.4)		
	Outpatient Department	6(20.0)	3(20.0)	3(20.0)		
	Others	9(30.0)	5(33.3)	4(26.6)		
Period of postgraduate degree	≤1	6(20.0)	4(26.7)	2(13.3)	2.50	0.476
	>1~≤5	7(23.3)	2(13.3)	5(33.3)		
	>5~<10	10(33.4)	6(40.0)	4(26.7)		
	≥10	7(23.3)	3(20.0)	4(26.7)		
	Mean ± SD	3.69 ± 4.75	6.11 ± 5.38	6.47 ± 4.20	−0.21	0.837
RPA		3.29 ± 0.52	3.33 ± 0.51	3.25 ± 0.55	0.44	0.662
EBPB		3.49 ± 0.422	3.49 ± 0.32	3.49 ± 0.51	0.03	0.979
EBPA		3.97 ± 0.34	3.90 ± 0.36	4.04 ± 0.31	−1.19	0.245
NPVS		3.45 ± 0.60	3.57 ± 0.65	3.33 ± 0.55	1.10	0.281

RPA = Research Practice Ability. EBPB = Evidence-Based Practice Beliefs. EBPA = Evidence-Based Practice Attitudes. NPVS = Nursing Professional Value Scale. SD = Standard Deviation. ICU = Intensive Care Unit.

**Table 2 healthcare-13-02273-t002:** Comparison of research variables between the two groups.

Variables	Experimental Group (n = 15)				Control Group (n = 15)				Experimental vs. Control Groups	
	Pre-Test (M ± SD)	Post-Test (M ± SD)	t/Z (*p*)	ES (95% CI)	Pre-Test (M ± SD)	Post-Test (M ± SD)	t (*p*)	ES (95% CI)	Post-Test Only t (*p*)	ES (95% CI)
RPA	3.33 ± 0.51	3.83 ± 0.70	−1.96 (0.050)	0.82 0.07–1.56)	3.25 ± 0.55	3.19 ± 0.54	−0.36 (0.721)	−0.11 (−0.83–0.61)	−2.39 (0.016)	1.02 (0.26–1.79)
EBPB	3.49 ± 0.33	3.84 ± 0.46	−3.30 (0.005)	0.87 (0.13–1.62)	3.49 ± 0.51	3.60 ± 0.47	−1.29 (0.219)	0.22 (−0.49–0.94)	1.37 (0.180)	0.52 (−0.21–1.24)
EBPA	3.90 ± 0.36	4.00 ± 0.41	−1.14 (0.275)	0.26 (−0.46–0.98)	4.04 ± 0.31	3.85 ± 0.25	1.54 (0.145)	−0.67 (−1.41–0.06)	1.16 (0.256)	0.44 (−0.28–1.17)
NPVS	3.57 ± 0.65	3.78 ± 0.62	−2.02 (0.063)	0.33 (−0.39–1.05)	3.33 ± 0.55	3.25 ± 0.60	0.69 (0.505)	−0.14 (−0.86–0.58)	2.38 (0.024)	0.87 (0.12–1.62)

Wilcoxon Signed-Rank Test. RPA = Research Practice Ability. EBPB = Evidence-Based Practice Beliefs. EBPA = Evidence-Based Practice Attitudes. NPVS = Nursing Professional Value Scale. SD = Standard Deviation. ES = Effect Size. CI = Confidence Interval.

## Data Availability

The data presented in this study are available upon reasonable request from the corresponding author. The data are not publicly available due to privacy and ethical restrictions.

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
