# Peer review of "Effectiveness of the Research Practice Ability Enhancement Program on Evidence-Based Practice Competencies in Clinical Nurses: A Non-Randomized Controlled Trial"

_healthcare, 2025, doi:10.3390/healthcare13182273_

Round 1

Reviewer 1 Report

Comments and Suggestions for Authors

Comments on the Manuscript Titled: "The Impact of a Research Practice Ability Enhancement Program for Clinical Nurses Led by Clinical Nurse Educator: A Mixed-Method Study"**

I have reviewed the study, and I find it interesting. The comments provided are important for improving this manuscript and could make it more valuable.

I am still unsure whether the full text of this study has been published previously; specifically, in reference 38, does it refer to the full text?

The methodology described within the abstract is not clear and needs clarification.

The translation into English seems to have been assisted by ChatGPT, and it requires revision for better accuracy and fluency.

The validity, reliability, CVI, and CVR of the tools should be explicitly stated.

Please explain why qualitative methods and focus groups were used, considering that a systematic review could also potentially provide similar insights.

In the problem statement section, expand on the increasing role of nurses, referencing the study "Identifying the fields of activity, challenges, and positive outcomes of nursing prescription: A systematic review." Given the importance of nurses' roles, emphasizing the need for RPAEP (Research Practice Ability Enhancement Program) is crucial.

In the qualitative section, only "content analysis" is mentioned, but the stages of the analysis are not specified. Please elaborate in more detail on the qualitative analysis process. The interviews were transformed into text, which was coded—please clarify the basis for the coding process.

The focus group method has not been explained.

How were categories and subcategories derived? Provide details on that process.

The discussion should be written more cautiously, avoiding overgeneralization of the results. The findings have limited generalizability.

At the end of the discussion, mention limitations such as the small sample size, the context-specific nature of the study, and cultural constraints.

One of the strengths of this study is its focus on a very important topic in nursing. Efforts should be made to shift nurses' perspectives toward this area, and more emphasis on this point is recommended.

In the results section, references should not be included.

If this study has been published previously or if part of its findings have been reported elsewhere, this should be clearly stated at the beginning of the methods section.

In the conclusion, note that these results are based on a single hospital in Korea, and overgeneralization should be avoided.

Comments on the Quality of English Language

The translation into English seems to have been assisted by ChatGPT, and it requires revision for better accuracy and fluency.

Author Response

All detailed responses to the reviewers’ comments have been provided in the attached response files.

Reviewer 2 Report

Comments and Suggestions for Authors

I would like to start by congratulating the authors on this thoughtful and well-structured study. The use of a mixed-methods approach adds depth to the findings.

That said, there are several methodological aspects that could benefit from further consideration. First, the study was conducted in a single university hospital, which limits the generalizability of the results. Additionally, using convenience sampling for both the experimental and control groups introduces a risk of selection bias and may reduce the representativeness of the sample. 

The power analysis is based on an exceptionally large effect size (d = 1.29), which substantially lowers the required sample size. While this may be statistically acceptable, it raises questions about the realism of such an assumption. With only 30 participants, the study is adequately powered only if the true effect is indeed very large; otherwise, the risk of a Type II error is considerable.

The qualitative component, comprising three focus groups with four participants each, is a valuable addition. However, the small sample size and lack of clarity regarding participant selection—whether random or voluntary—may affect the reliability and saturation of the data. It would be helpful to know whether the themes identified were consistent across groups and whether data saturation was assessed.

Moreover, the manuscript does not report effect sizes or confidence intervals, which are essential for interpreting the magnitude and precision of the results. A p-value–centric approach may obscure the practical significance of the findings.

The intervention itself appears resource-intensive, requiring mentorship, time, and institutional support. This raises questions about its scalability and feasibility in settings with limited resources.

Finally, I would encourage the authors to clarify the rationale for including master’s students as participants. Their prior exposure to research training could influence the outcomes and should be acknowledged as a potential confounding factor.

With some refinements and clarifications, this study has the potential to make a meaningful contribution to nursing education and professional development.

Author Response

(The authors gave the same response as above.)

Reviewer 3 Report

Comments and Suggestions for Authors

Thank you for inviting me to review this manuscript reporting a trial of an intervention to strengthen evidenced based practice by clinical nurses. The concept of this study is good as is the intervention, but the reporting of this study needs to be amended to align with the consort guidelines. This is pitched as a quasi-experiment, which is an out-of-date term – this is a non-randomised controlled trial. I offer the following suggestions to strengthen its reporting:

  1. This was a trial, the authors might have used mixed methods as the method of evaluation, but it was first and foremost a trial. The title needs to change to reflect this – use the PICO framework as guide to formulating the title.
  2. Abstract: this needs re-writing and should follow the consort abstract guidance.
  3. Page 2, first paragraph contains sentences repeated in the paragraph above.
  4. It would benefit the paper if the study hypotheses were presented so it can be clear what is the primary outcome and secondary outcomes.
  5. Sample size calculations are based on the effect size, and this usually comes from the primary outcome measure results of previous papers. The authors need to give the rationale and reference for the effect size they propose; 24 participants for a trial does not seem a lot.
  6. The outcome measures need to be presented as the primary and secondary to align with the hypotheses.
  7. As suggested in the consort guidelines, details of the intervention need to be provided in enough detail for this to be replicated. It also needs to be made clear what the control group did – it reads as if they completed the baseline questionnaire then the end of study questionnaire, when they had access to details of the intervention. They then took part in the intervention – this sounds like a waitlist control trial – did the authors measure impact of the control group when they had finished their participation?
  8. The recruitment of nurses is explained but it’s unclear how they were allocated to the intervention group. Was this randomised or pure convenience?
  9. The qualitative aspect, it needs to be specified that this only included nurses who were in the intervention arm. The focus groups (not interviews, this is a superfluous word), how many were conducted and how many were in each group.
  10. Terminology to explain who was involved in various aspects is confusing – the educators are referred to as researchers and then there is a researcher conducting the focus groups. Are these the same people (which would introduce a huge amount of bias) – this needs to be clearer.
  11. Table 1 needs to have ‘baseline’ included in the title.
  12. Did all nurses remain in the study and complete all the measures?
  13. The qualitative aspect of the study needs to be reported as a qualitative study – healthcare does not have a restriction on words so there is no need to tabulate quotes. There needs to be a rich narrative to provide an explanation around the theme title, supported with the most relevant quotes. A table of quotes is not particularly enlightening or interesting.
  14. Page 12, third paragraph has the abbreviation FGI – I can see this in the abbreviation list but it is unnecessary and the authors can just put focus group.

Author Response

(The authors gave the same response as above.)

Reviewer 4 Report

Comments and Suggestions for Authors

Thank you for the opportunity to review this rigorous examination of a program to enhance nurse capability in evidence based practice.  The Research Practice Ability Enhancement Program (RPAEP) supports nurses to examine the peer-reviewed literature and research methodologies which enhance their confidence to identify as, and positive attitude towards research-informed and evidence-based practice.

This well crafted mixed-methods, quasi-experimental study included 30 participating nurses from one teaching hospital, 15 participants in each of the intervention and control groups.  The program was run over 12 sessions. Qualitative data was collected in focus groups with 12 participants.

Previously designed and validated scales were used to assess participant pre-post:  beliefs; attitudes; Nursing professionalism; and  Research practice ability.

Findings

Participant confidence and positive attitudes towards research improved in the intervention group compared to the control group. The qualitative component allowed nuanced interpretation of the influence of systemic and organisational context on participant capacity to enact research and evidence-based practice. 

Limitations

At line 384 the study participants are referred to as 'subjects'. Please use the word 'participants' to better acknowledge their autonomy and informed consent to take part in the study.

The authors acknowledge that larger studies may be needed.

The only suggestion for improvement is to consider how this may be scaled and expanded in the current Korean hospital system. Can you make comment on some of the key authorising policy measures (pay, recognition, time allocated) or context (relational, staffing levels...).  And can you suggest how modern AI and digital platforms might facilitate this process.  Are their recommendations to make to nurse educators?

On the whole, this is a well written and very neat study which has learnings relevant to other human services. 

Author Response

(The authors gave the same response as above.)

Round 2

Reviewer 3 Report

Comments and Suggestions for Authors

Thank you for address my comments. I am happy with all the changes you have made. I have noticed one minor correction. In the limitations you put: "However, several 
methodological limitations remain." This sentence has been included twice so one needs removing.

Author Response

Thank you for your careful review and helpful suggestion. We have revised the Limitations section by removing the duplicated sentence (“However, several methodological limitations remain”). Now the text only retains the sentence “Several methodological limitations should be acknowledged” to avoid redundancy.
